# CAN TRUSTWORTHINESS GENERALIZE? LEVERAGING WEAK SUPERVISION FOR STRONGER MODELS

## ABSTRACT

As large language models continue to advance, ensuring their trustworthiness is critical. However, inaccessible real-world ground truth labels pose a significant challenge in high-stakes domains. Recent studies have highlighted weak-to-strong generalization, where a strong model trained only on a weak model's labels surpasses the weak model in task performance. Yet, whether critical trustworthiness properties such as robustness, fairness, and privacy can generalize similarly remains an open question. This is the first work to study this question by examining if a stronger model can enhance trustworthiness when fine-tuned on a weaker model's labels, a paradigm we term *weak-to-strong trustworthiness*. To address this, we introduce two fundamental fine-tuning strategies that leverage trustworthiness regularization during the fine-tuning of the weak model and the weak-to-strong transfer. Our experimental evaluation on real-world datasets reveals that while some trustworthiness properties, such as fairness, adversarial robustness, and OOD robustness, show significant improvement in trustworthiness generalization when both models were regularized, others like privacy do not exhibit signs of weak-to-strong trustworthiness. Our results highlight the potential of weak-to-strong trustworthiness as a practical pathway for enhancing the trustworthiness of increasingly capable AI systems, even under imperfect real-world conditions.

## 1 INTRODUCTION

In recent years, developments in large language models (LLMs) have demonstrated breakthroughs in capability and scale (Radford et al., 2019; Bubeck et al., 2023). As models continue to improve, trustworthiness has emerged as a critical aspect of AI systems, especially as LLMs are increasingly deployed in high-stakes domains like healthcare, finance, and criminal justice (Wang et al., 2023).

A fundamental challenge in developing trustworthy models is that real-world supervision is often imperfect. The lack of ground-truth labeled data is a bottleneck for training capable models, particularly in the domains where trustworthiness matters most. For instance, in medical diagnosis, we may not always have perfect ground truth labels because even expert doctors can disagree about a patient's condition, or a definitive diagnosis might only be possible after invasive tests or autopsy. As a result, the labels used for training are noisy or incomplete, rather than perfect ground truth. The challenge of imperfect supervision parallels a question in AI alignment: if we only have access to potentially biased supervision (like imperfect human decisions), how can we control more capable AI systems to be more aligned with human values and trustworthiness?

A recent study demonstrated the phenomenon of weak-to-strong (WTS) generalization, where a strong model outperforms a weak model by fine-tuning on only the weak model's labels (Burns et al., 2024). Weak-to-strong learning is particularly promising for studying superalignment, where ground truth labels are unknown by humans, addressing the real-world inaccessibility of ground truth data (Bach et al., 2017; Ratner et al., 2017) (Figure 1). A few follow-up studies have focused on applying weak-to-strong learning to improve performance in various settings, yet none have investigated trustworthiness (Chen et al., 2024; Yang et al., 2024).

In this work, we introduce the *weak-to-strong trustworthiness* paradigm. We investigate the unexplored question: *Can trustworthiness properties be generalized to a strong model from fine-tuning on a weak model's labels?*

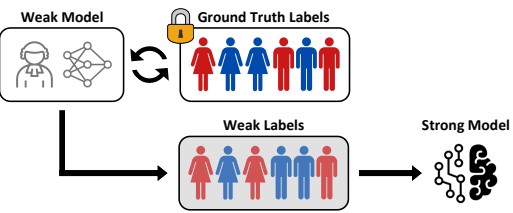

Figure 1: **Weak-to-strong framework for when ground truth labels are unavailable.** The weak model (e.g. human supervision or small LLM) has been trained to predict an inaccessible set of complete ground truth labels. The weak labels (weak model's predictions) are then used to fine-tune the strong model.

While previous work mainly use weak-to-strong learning to enhance raw predictive accuracy, our objective is to show that trustworthiness can also be improved when fully ground truth labels remain unavailable (Chen et al., 2024; Yang et al., 2024). In the context of the superalignment scenario, our approach examines if superintelligent strong models trained on human weak labels can overcome human biases to become more trustworthy.

To enable a systematic study of this phenomenon, we develop two fundamental fine-tuning strategies. We perform rigorous empirical experiments using the Pythia model suite (Biderman et al., 2023) to analyze our fine-tuning strategies on standard trustworthiness datasets. Our main contributions are:

- **Weak-to-strong trustworthiness is feasible:** We present the novel conceptual framework of weak-to-strong trustworthiness. As the first study examining whether trustworthiness properties generalize through WTS learning, our results indicate that *WTS trustworthiness is indeed feasible*.
- **Standard weak-to-strong learning is insufficient:** Standard fine-tuning a strong model on a weak model's labels yields inconsistent generalization of trustworthiness across properties (fairness, OOD robustness, adversarial robustness, privacy).
- **Fundamental fine-tuning strategies improve weak-to-strong trustworthiness**: We introduce the strategies Trustworthiness Fine-tuning (TFT), which regularizes weak model training, and Trustworthiness Fine-tuning and Transfer (TFTT), which regularizes both weak model training and weak-to-strong learning. TFTT consistently improves trustworthiness generalization, significantly enhancing fairness and robustness. Our strategies are summarized in Figure 2.
- **Comprehensive empirical evaluation**: We evaluate our strategies across 4 properties, 20 datasets, 14 definitions and tasks, and 5 model sizes ranging from 14M to 6.9B parameters. In addition, our sensitivity analysis demonstrates consistent weak-to-strong trustworthiness across a wide range of hyperparameter values.

Our study is critical for understanding the promising potential and limitations of weak-to-strong trustworthiness. Our findings have broad implications for the future of AI development: by demonstrating that trustworthiness properties can be systematically enhanced as models scale, we provide a pathway for ensuring that increasingly powerful AI systems remain aligned with human values even when perfect supervision is unavailable.

## 2 RELATED WORK

This work is the first to study trustworthiness generalization from a weak supervisor to a strong model. We discuss related works for the topics below.

**Fairness.** Unfair outcomes can arise in language models when they inadvertently encode biases present in the training data, leading to discriminatory practices against certain groups based on sensitive attributes like race, gender, or age (Bolukbasi et al., 2016). Recent efforts to improve fairness in LLMs include data pre-processing, post-processing, and adversarial training such as augmenting training data to balance gender representations (Zhao et al., 2018) and debiasing word embeddings (Huang et al., 2020). Our study is distinguished by its weak-to-strong setting and integration of fairness directly into the model's learning objective during fine-tuning.

**Out-of-distribution robustness.** OOD robustness describes a model's ability to perform well on inputs that differ from its training distribution. Various methods aim to enhance OOD robustness, including data augmentation techniques like adversarial perturbations (Madry et al., 2018; Lecuyer et al., 2019), EDA (Wei & Zou, 2019), as well as training modifications like label smoothing (Szegedy et al., 2016) and focal loss (Lin, 2017). However, recent research has shown that many

of these methods do not reliably improve OOD robustness and may even degrade performance on in-distribution tasks; standard fine-tuning often remains a strong baseline (Yuan et al., 2023). In this work, we employ adversarial perturbation as a representative robustness technique, which has been explored in existing LLM robustness literature (Zhu et al., 2019; Ye et al., 2023). Unlike prior approaches, we focus on generalizing OOD robustness from weak models to larger strong models, both with and without the use of robustness-enhancing regularization.

**Adversarial robustness.** Machine learning model outputs can be changed by introducing minimal perturbations to a benign input, causing the model to malfunction (Szegedy et al., 2014; Goodfellow et al., 2015; Madry et al., 2018). Existing adversarial attack algorithms have been shown to degrade a large language model's performance on natural language processing tasks such as sentiment analysis, question answering, text classification, and entailment (Jin et al., 2020; Zang et al., 2020; Wang et al., 2020; Li et al., 2020; Garg & Ramakrishnan, 2020). Our work differs from these existing studies and is the first to examine if adversarial robustness can generalize from a weak model to a larger strong model fine-tuned on weak labels.

**Privacy and model distillation.** Prior research has explored knowledge distillation as a mechanism to mitigate privacy attacks. One example is the PATE framework (Papernot et al., 2016), where knowledge distillation is employed to reduce an ensemble of teacher models into a single model with provable privacy guarantees (Dwork et al., 2006). Other works have built on this idea, such as Zheng et al. (2021) and Tang et al. (2022), to similarly construct privacy-preserving model ensembles and consolidate them through distillation. Some research suggests that distillation alone can serve as an effective privacy defense (Shejwalkar & Houmansadr, 2021). Building on this, Mazzone et al. (2022) investigate the use of repeated distillation to protect against membership inference attacks. However, Jagielski et al. (2024) demonstrate through privacy attacks that distilled models without privacy guarantees can still leak sensitive information. In contrast to prior work, our research focuses on the privacy implications of weak-to-strong learning. This approach is the inverse of traditional model distillation. Nothing is known about the privacy risks when this process is reversed, making our work an important contribution to the field.

# 3 OUR FRAMEWORK

In Section 3.1, we discuss how we adapt the weak-to-strong learning framework introduced by Burns et al. (2024) for trustworthiness. Then, we introduce our fine-tuning strategies for studying weak-to-strong trustworthiness in Section 3.2. Afterwards, in Section 3.3, we describe regularization strategies to enhance trustworthiness properties such as fairness, robustness, and privacy.

## 3.1 PRELIMINARIES

**Notation.** We consider training datasets of the form $\{(x_i, y_i)\}_{i=1}^N$ where $y_i \in \mathcal{Y}$ is the ground-truth label. We denote a classifier $f_\theta : \mathcal{X} \to \mathcal{Y}$ parametrized by $\theta \in \mathbb{R}^d$, mapping inputs $x \in \mathcal{X}$, to labels $\mathcal{Y}$. We define the outputs of a fine-tuned smaller classifier $f_w(x)$ as *weak labels*, where $w \in \mathbb{R}^k$ denotes a lower-capacity parameterization than $\theta$ where $k \ll d$. Let $\ell : \mathbb{R} \times \mathbb{R} \to \mathbb{R}$ represent an appropriate loss function such as cross-entropy loss.

**Weak-to-strong learning.** In the weak-to-strong (WTS) framework, a pre-trained strong model accomplishes performance generalization by fine-tuning on a weak model's labels. Burns et al. (2024) defined two methods for the WTS transfer: WTS-Naive and WTS-Aux-Loss. WTS-Naive refers to the strong model doing regular fine-tuning on weak labels. WTS-Aux-Loss consists of an additional auxiliary loss, weighted by $\alpha \in [0, 1]$ to adjust the confidence in the strong model's predictions relative to the weak labels. This auxiliary loss encourages the strong model to make confident predictions, even when they diverge from the weak labels, potentially enhancing generalization. WTS-Naive is equivalent to setting $\alpha = 0$. WTS-Aux-Loss refers to when $\alpha > 0$.

Similarly to Burns et al. (2024), our loss function is a linear combination of the cross-entropy losses from the weak and strong models. However, we incorporate trustworthiness regularization ($\lambda$):

$$\ell_{\text{WTS}}^{\text{AUX}} = (1-\alpha)\ell\big(f_\theta(x), f_w(x;\lambda)\big) + \alpha\ell\big(f_\theta(x;\lambda), \hat{f}_{t,\theta}(x)\big). \tag{1}$$

$f_w(x;\lambda)$ denotes the weak model fine-tuned with trustworthiness regularization strength $\lambda$ and $f_\theta(x)$ denotes the strong model. Further, $\hat{f}_{t,\theta}(x)$ represents the hardened strong model predictions accord-

ing to threshold $t$ set proportional to the dataset class weights. When $\lambda = 0$, we are in the standard WTS setting studied by Burns et al. (2024) (No TFT). In our proposed strategies, we apply regularization with $\lambda > 0$ to the weak model (TFT, TFTT) and the weak-to-strong learning (TFTT).

We distinguish between the baseline WTS transfer methods (WTS-Naive and WTS-Aux-Loss), which are the core learning algorithms, and our overarching fine-tuning strategies (TFT and TFTT), which are the experimental frameworks that employ them. While the transfer methods are the specific algorithms used to train the strong model on weak labels, our strategies augment this process by incorporating trustworthiness regularization. In our experiments, each strategy is evaluated using both WTS-Naive and WTS-Aux-Loss for the transfer step.

## 3.2 Fine-tuning Strategies for Studying Weak-to-Strong Trustworthiness

We systematically study how trustworthiness generalizes by applying regularization at two key stages: (1) the initial fine-tuning of the weak model on ground truth data, and (2) the subsequent weak-to-strong (WTS) transfer to the strong model. This results in three distinct strategies, each incorporating a greater degree of regularization (Figure 2).

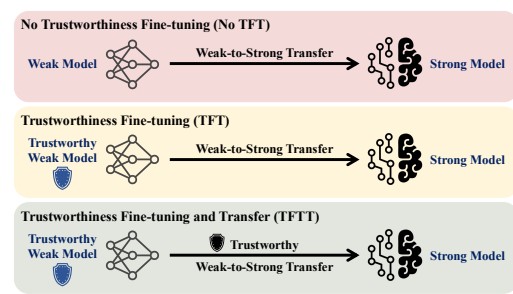

Figure 2: **Our fine-tuning strategies**.

The baseline strategy, **No Trustworthiness Fine-tuning (No TFT)**, follows Burns et al. (2024) and applies no regularization at either stage ($\lambda = 0$). A standard weak model is trained, and its labels are used for a standard WTS transfer.

Our first proposed strategy, **Trustworthiness Fine-tuning (TFT)**, introduces regularization to the first stage only. We fine-tune the weak model with a trustworthiness objective ($\lambda > 0$) to create a *trustworthy weak model*. This model's labels are then used in a standard WTS transfer.

Our second and most comprehensive strategy, **Trustworthiness Fine-tuning and Transfer (TFTT)**, applies regularization at both stages. It uses the same trustworthy weak model from TFT, but then additionally incorporates a trustworthiness objective into the WTS transfer process itself. We detail this *trustworthy weak-to-strong transfer* objective in Appendix A.1.

## 3.3 Regularization for Enhancing Trustworthiness Properties

We enhance trustworthiness by incorporating one of the following regularization techniques during fine-tuning.

**Fairness.** We enforce statistical fairness criteria such as Demographic Parity, which requires that the prediction rates be equal across groups based on a protected attribute $a$ (e.g., gender). Following Zafar et al. (2017), we use an objective that balances the standard loss with a regularization term that minimizes the covariance between model outputs $f_w(x_i)$ and the sensitive attribute $a_i$:

$$\min_w \frac{1}{N} \sum_{i=1}^{N} \ell(f_w(x_i), y_i) + \lambda_{\text{Fair}}(a_i - \bar{a})f_w(x_i), \tag{2}$$

where $\bar{a}$ is the base rate of the attribute. The hyperparameter $\lambda_{\text{Fair}}$ controls the accuracy-fairness trade-off. We use a similar objective to enforce Equalized Odds.

**Adversarial Robustness.** To improve robustness against adversarial attacks, we train the model on both clean and adversarially perturbed samples. The training objective is a weighted average of the loss on the original inputs $x_i$ and their adversarial counterparts $x_i'$:

$$\min_w \frac{1}{N} \sum_{i=1}^{N} (1 - \lambda_{\text{Adv}})\ell(f_w(x_i), y_i) + \lambda_{\text{Adv}}\ell(f_w(x_i'), y_i), \tag{3}$$

where $\lambda_{\text{Adv}}$ controls the emphasis on robustness.

**Out-of-Distribution (OOD) Robustness.** Following prior work (Madry et al., 2018; Lecuyer et al., 2019; Zhu et al., 2019; Bowman et al., 2015; Li et al., 2019), we enhance OOD robustness by adding Gaussian noise $z \sim \mathcal{N}(0, \lambda_{\text{OOD}} \cdot \text{I}_d)$ to the word embeddings $e(x)$ before they are processed by the model. The objective is the standard cross-entropy loss on these noisy inputs:

$$\min_w \frac{1}{N} \sum_{i=1}^{N} \ell\big(y_i, f_w(x_i; \lambda_{\text{OOD}}))\big), \tag{4}$$

where $\lambda_{\text{OOD}}$ controls the noise variance and thus the strength of the regularization.

**Privacy.** We use Differentially Private SGD (DP-SGD) (Abadi et al., 2016) to provide formal $(\lambda_P, \delta)$-differential privacy guarantees. An algorithm is differentially private if its output is nearly identical whether or not any single data point is included in its training set:

$$\mathbb{P}(\mathcal{A}(D_1) \in S) \leq \exp(\lambda_P) \cdot \mathbb{P}(\mathcal{A}(D_2) \in S) + \delta. \tag{5}$$

DP-SGD achieves this by modifying the standard training process: for each batch, it computes per-sample gradients, clips their L2 norm to a constant $C$, aggregates them, and adds calibrated Gaussian noise before applying the update. The amount of noise is tuned for the desired privacy level $(\lambda_P, \delta)$.

# 4 EXPERIMENTAL EVALUATION

In Section 4.1, we empirically evaluate weak-to-strong trustworthiness using the three fine-tuning strategies discussed in Section 3. Then, in Sections 4.3 and Appendix C, we perform a comprehensive sensitivity analysis, varying the model size, regularization strength, and other hyperparameters specific to weak-to-strong learning. We begin by describing the real-world datasets used in our experiments, followed by an overview of the models and the strong ceiling upper bounds we use. Table 3 provides an overview of all properties, metrics, datasets, and tasks.

**Datasets.** We evaluate trustworthiness generalization using 20 datasets, including the Enron Email dataset (Klimt & Yang, 2004), the AG News dataset, the Adult dataset (Ding et al., 2021), the PUMS ACS dataset (Ding et al., 2021), the OOD Style Transfer datasets (Wang et al., 2023), and the AdvGLUE++ datasets (Wang et al., 2023). For all datasets, we show average results from multiple runs and report $\pm 1$ standard deviation. While the main paper's plots focus on Enron, Adult, OOD Style Transfer, and AdvGlue++ datasets, supporting results on the other datasets can be found in Appendix C. Additional dataset and experimental details are in Appendix D.

**Large language models.** We fine-tune models from the Pythia suite spanning five model sizes: 14M, 70M, 410M, 1B, 6.9B parameters (Biderman et al., 2023). The wide range of sizes allows us to systematically explore how model size impacts weak-to-strong trustworthiness.

**Metrics.** We evaluate a model's trustworthiness as follows:

- **Fairness**: We evaluate fairness using the demographic parity and equalized odds. For both definitions, lower values indicate better fairness, as they reflect minimal disparity in predictions between protected groups. We conduct comprehensive experiments using Demographic Parity Difference (DPD), defined as DPD $= \mathbb{P}(f_\theta(x) = 1|a = 1) - \mathbb{P}(f_\theta(x) = 1|a = 0)$. Additional experiments on Equalized Odds Difference support the trends observed (Figure A12).

- **Robustness**: For robustness, we measure both OOD accuracy and adversarial accuracy, abbreviated as Robust Accuracy (RA), by evaluating the model's performance on OOD and adversarially perturbed test data. Specifically, we compute the RA $= \frac{1}{n_{\text{test}}} \sum_{i=1}^{n_{\text{test}}} \mathbb{I}[f_\theta(x_i') = y_i]$, where $x'$ represents either an OOD sample or an adversarially perturbed input, and $\mathbb{I}$ denotes the indicator function that equals 1 if the prediction is correct.

- **Privacy**: We evaluate privacy using targeted data extraction attacks and membership inference attacks (Shokri et al., 2017; Carlini et al., 2021). We conduct comprehensive experiments using extraction attacks, where given a prefix sequence and a generated response of $k$ tokens, we compute the extraction rate by determining what fraction of the $k$-token continuation (suffix) matches the ground truth continuation of the sample. A higher extraction rate indicates a greater risk that the model memorizes and extracts private information. Additional experiments on standard membership inference attacks support the trends observed (Figure A13b).

**Strong ceiling upper bound.** For comparison, we establish upper bounds for a strong model's trustworthiness by fine-tuning it using ground truth labels and regularization. This value, referred to as the *strong ceiling*, represents the strong model utilizing its full capabilities. Section A.2 provides more details on determining the strong ceiling.

## 4.1 EVALUATING WEAK-TO-STRONG TRUSTWORTHINESS

We define weak-to-strong trustworthiness as a monotonic trend – starting from the weak model and increasing through the WTS-Naive and WTS-Aux-Loss models, with the strong ceiling as the upper bound. Despite only fine-tuning on the weak model' labels, the strong model is able to generalize trustworthiness and recover part of the trustworthiness gap from the weak model to the strong ceiling upper bound.

We present our results for all four trustworthiness properties across the three strategies in Table 1, and throughout Figures 3-6. Figure A2 shows the properties across all three strategies side-by-side.

**No TFT.** The No TFT fine-tuning strategy does not achieve consistent weak-to-strong trustworthiness (Figure 3). For fairness experiments, the level of unfairness (demographic parity difference) remains constant at around 35% across all weak and strong models. Similarly, we do not observe privacy generalization (Figure 6). We expected no consistent weak-to-strong trustworthiness for No TFT (standard weak-to-strong) as the strategy lacks regularization to explicitly enforce trustworthiness. Surprisingly, we observe a weak-to-strong trustworthiness trend for OOD and adversarial robustness. Despite the absence of regularization, the WTS-Naive and WTS-Aux-Loss models exhibited improved robustness compared to the weak models, suggesting that some trustworthiness properties may naturally generalize without explicit constraints.

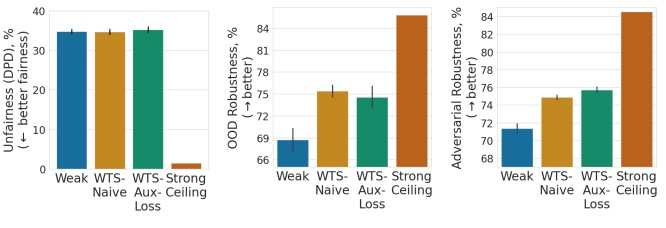

(a) Fairness  (b) OOD Robustness  (c) Adv. Robustness

Figure 3: **No TFT (standard weak-to-strong) is insufficient for trustworthiness generalization.** Weak-to-strong trustworthiness is inconsistent across properties, from no generalization of fairness to generalization of OOD and adversarial robustness.

**TFT.** The TFT fine-tuning strategy significantly improves the trustworthiness of weak models across all four properties (Figures 4, 6). The effect of the additional regularization applied to weak models aligns with our expectations, as weak models are now explicitly regularized to enhance trustworthiness. Compared to No TFT, the weak models achieve lower unfairness (5% from 35%), increased OOD robustness (72% from 69%), increased adversarial robustness (78% from 71%), and lower privacy extraction (15% from 19%). Despite the trustworthy weak models, TFT does not achieve consistent weak-to-strong trustworthiness. We only observe generalization for OOD robustness (Figure 4b). The strong models are not more trustworthy than the weak models for fairness, adversarial robustness, and privacy (Figures 4a, 4c, 6).

**TFTT.** The TFTT fine-tuning strategy significantly improves the trustworthiness of strong models across all four properties (Figures 5, 6). The effect of the additional regularization applied to weak and strong models aligns with our expectations, as both models are now explicitly regularized to enhance trustworthiness. Compared to No TFT, the strong models achieve lower unfairness (2%

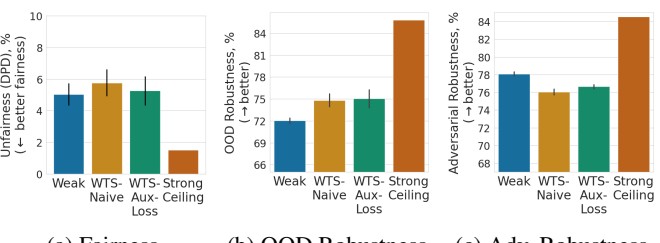

(a) Fairness  (b) OOD Robustness  (c) Adv. Robustness

Figure 4: **TFT improves trustworthiness of weak models.** However, weak-to-strong trustworthiness is still inconsistent across properties, from no generalization of fairness and adversarial robustness to generalization of OOD robustness.

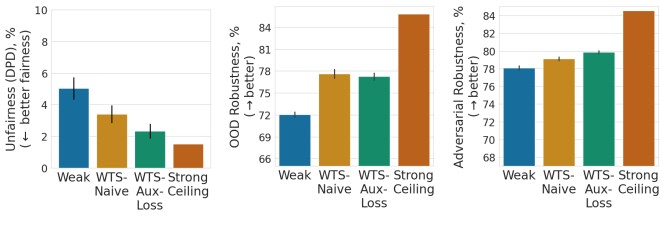

(a) Fairness     (b) OOD Robustness     (c) Adv. Robustness

Figure 5: **TFTT achieves consistent WTS trustworthiness.** TFTT significantly improves trustworthiness generalization for fairness, OOD robustness, and adversarial robustness.

from 35%), increased OOD robustness (78% from 75%), increased adversarial robustness (80% from 75%), and lower privacy extraction (26% from 45%).

Unlike previous strategies, TFTT achieves consistent weak-to-strong trustworthiness for fairness, OOD robustness, and adversarial robustness (Figure 4a). The strong models are significantly more trustworthy than the weak models, indicating successful trustworthy generalization through TFTT. For fairness and adversarial robustness, the WTS-Aux-Loss models generalize more effectively than the WTS-Naive models, suggesting that the auxiliary loss enables more weak-to-strong trustworthiness.

Through TFTT, strong models are able to recover a significant portion of the trustworthiness gap between the weak model and strong ceiling. Despite their lack of ground truth labels, strong models recover 88% of the fairness gap (2.8% out of 3.2%), 41% of the OOD robustness gap (5.5% out of 13.5%), and 31% of the adversarial robustness gap (2% out of 6.5%) (Figure 5).

**Trade-off between trustworthiness and task performance.** For fairness and adversarial robustness, weak-to-strong trustworthiness includes a slight decline in task performance (Figure A2). However, the performance decrease does not exceed 1% from weak to strong models while trustworthiness generalized to recover up to 88% of the trustworthiness gap. Our results demonstrate that significant trustworthiness generalization can be achieved with minimal impact on task performance.

| | Fairness | OOD Robustness | Adv. Robustness | Privacy |
|---|---|---|---|---|
| **No TFT** | ✗ | ✓ | ✓ | ✗ |
| **TFT** | ✗ | ✓ | ✗ | ✗ |
| **TFTT** | ✓ | ✓ | ✓ | ✗ |

Table 1: **Weak-to-strong trustworthiness across properties and fine-tuning strategies.** TFTT achieves consistent weak-to-strong trustworthiness in fairness, OOD robustness, and adversarial robustness.

## 4.2 PROPERTY-SPECIFIC GENERALIZATION BEHAVIORS

**Privacy.** As the only property to not demonstrate consistent weak-to-strong trustworthiness under the TFTT strategy, privacy presents a unique situation. However, note that the strong ceiling does not achieve better privacy than the weak model, which prevents any monotonic weak-to-strong privacy trend.

One reason for this distinction is that privacy is measured with respect to the underlying training dataset (see Appendix D). Larger models are more capable of memorizing information, leading to a greater risk of private information leakage (Lee-mann et al., 2024). As a result, larger mod-

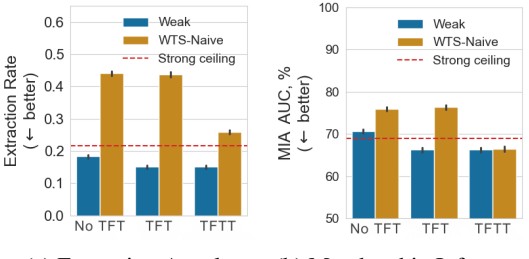

(a) Extraction Attack     (b) Membership Inference

Figure 6: **No weak-to-strong privacy.** While TFTT does not achieve privacy generalization, it still improves the privacy of strong models compared to other strategies.

els are more susceptible to leaking private data than smaller models. Therefore, we observe that privacy, measured by the extraction rate or membership inference attack success in Figure 6, degrades when learning from a weak model to a strong model. This inherent relationship between model

capacity and memorization makes privacy fundamentally different from properties like robustness, where a larger model's capacity can be leveraged to *improve* the property rather than degrade it.

**Robustness and Fairness.** Unlike privacy, which fails to generalize, and fairness, which requires full regularization via TFTT, OOD and adversarial robustness exhibit a tendency to generalize even in the baseline No TFT setting. One possible explanation is that robustness is a more localized property tied to the model's feature representations and the geometry of its decision boundary in the immediate vicinity of an input (Madry et al., 2018). A strong model, with its higher capacity, can learn a smoother and more effective representation even from the noisy labels of a weak supervisor, leading to these natural improvements.

In contrast, fairness is a global statistical property of the output distribution across entire protected groups (Zafar et al., 2017). This aggregate statistical constraint may be too complex to infer from weak labels alone, requiring explicit regularization during the WTS transfer to be enforced effectively.

### 4.3 SENSITIVITY ANALYSIS

We conduct a comprehensive sensitivity analysis to explore how various parameter values influence trustworthiness generalization. Specifically, we examine the impact of model size and regularization strength ($\lambda$). This analysis validates the robustness of our main results and demonstrates the conditions under which our strategies are most effective. Further analysis on the auxiliary loss parameter ($\alpha$) can be found in Appendix C.

**Impact of Model Size.** Our analysis reveals that the trustworthiness generalization trends hold consistently across all five weak/strong model configurations we tested (see Figures A7-A11). Beyond this consistency, we find that the capacities of the weak and strong models play distinct and important roles. While increasing the strong model size led to some trustworthiness improvements, we saw significant improvement in weak-to-strong trustworthiness after increasing the weak model size

As illustrated with OOD robustness in Figure 7, increasing the **weak model's** capacity (e.g., from 14M to 70M parameters, comparing Fig. 7a and 7b) boosts the trustworthiness of *both* the weak supervisor and the resulting strong model. In contrast, increasing the **strong model's** capacity (e.g., from 410M to 6.9B, comparing Fig. 7a and 7c) primarily improves the generalization of the strong model itself. This suggests that while a more capable strong model is better at learning, starting with a more capable weak supervisor provides a better foundation for the entire WTS trustworthiness pipeline.

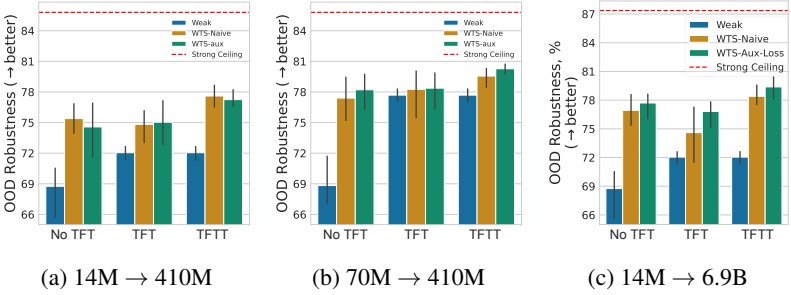

|       (a) 14M $\rightarrow$ 410M       |       (b) 70M $\rightarrow$ 410M       |       (c) 14M $\rightarrow$ 6.9B       |

Figure 7: **Impact of model size on OOD Robustness.** Increasing the weak model's size improves trustworthiness for both weak and strong models (7a vs. 7b), while increasing the strong model's size primarily enhances the strong model's generalization (7a vs. 7c).

**Impact of Regularization Strength** ($\lambda$). Our TFTT strategy demonstrates robustness to the specific choice of the regularization strength, $\lambda$. As shown in Figure 8, for fairness, OOD robustness, and adversarial robustness, the monotonic trend of WTS trustworthiness is maintained across a wide range of $\lambda$ values. This indicates that the effectiveness of TFTT is not reliant on fragile hyperparameter tuning, making it a practical and reliable method. This stability contrasts with the TFT strategy, where the final outcome is more sensitive to the initial regularization of the weak model (Figure A3). Depending on the regularization strength and property, the strong model's improvement over

the weak model is not as significant (OOD robustness), or the strong model may not improve over the weak model at all (fairness, adversarial robustness).

The success of TFTT across various $\lambda$ values confirms our central finding: applying regularization during the WTS transfer itself is the key to achieving significant and consistent trustworthiness generalization.

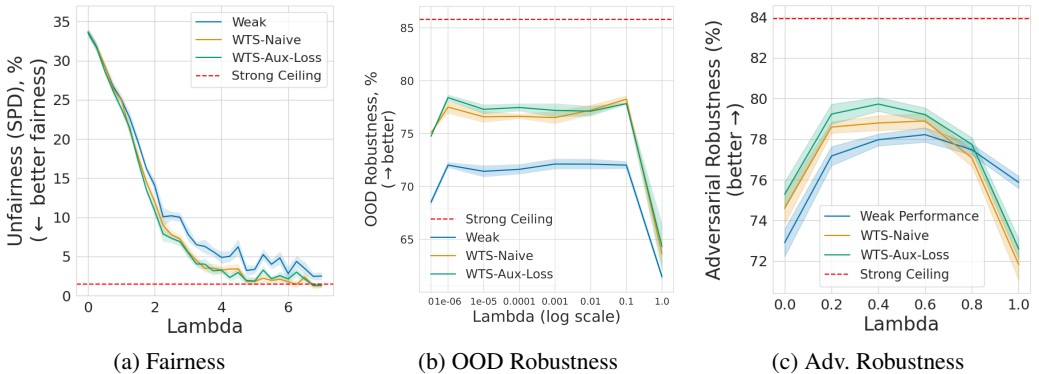

(a) Fairness      (b) OOD Robustness      (c) Adv. Robustness

Figure 8: **TFTT consistently improves trustworthiness across regularization strengths ($\lambda$).** The monotonic WTS trend (Weak < WTS-Naive/Aux-Loss < Strong Ceiling) holds for a wide range of $\lambda$ values.

## 5 CONCLUSION

Our work provides the first systematic investigation into whether critical trustworthiness properties like fairness, robustness, and privacy can be generalized through weak-to-strong learning in language models. We term this process weak-to-strong trustworthiness. Based on our novel conceptual framework, we make several key contributions.

First, we show that standard weak-to-strong learning alone is insufficient for consistent trustworthiness generalization, underlining the need for integrating regularization in weak-to-strong learning. Consequently, we introduce two fundamental fine-tuning strategies, TFT and TFTT, that significantly improve the trustworthiness of weak labels and achieve consistent weak-to-strong trustworthiness. Our TFTT strategy, in particular, demonstrates remarkable success in recovering up to 88% of the trustworthiness gap between weak models and strong ceiling baselines, while simultaneously maintaining strong task performance. While our results show consistent weak-to-strong trustworthiness for properties like fairness and robustness, the distinct behavior we observed with privacy generalization highlights the nuanced and property-specific nature of trustworthiness transfer in language models.

Our findings have broad implications for the development of trustworthy AI systems. By demonstrating that trustworthiness properties can be systematically enhanced through our proposed strategies, we provide a practical pathway for ensuring increasingly powerful models remain aligned with human values - even in real-world settings with inaccessible ground truth labels. As AI systems continue to grow in capability and autonomy, ensuring that trustworthiness generalize without requiring perfect supervision will be crucial for their safe deployment in high-stakes domains.

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

# A WEAK TO STRONG LEARNING PROCESS

## A.1 TRAINING OBJECTIVE FOR TFTT

In this section, we give a detailed description of the loss used for the third fine-tuning strategy presented in Section 3.2.

**Fairness.** To incorporate the fairness constraint into the fine-tuning process, we apply regularization twice yielding the following objective

$$
\theta^* \in \arg\min_{\theta} \mathcal{L}_{\text{Fair}}^{\text{WTS}}(\theta; \lambda_{\text{Fair}}^{\text{W}}, \lambda_{\text{Fair}}^{\text{WTS}}, \alpha, f_w)
$$

$$
= \arg\min_{\theta} \frac{1}{N} \sum_{i=1}^{N} \ell_{\text{WTS-AUX}}(x_i, f_\theta; \alpha, \lambda_{\text{Fair}}^{\text{W}}, f_w) + \lambda_{\text{Fair}}^{\text{WTS}} \cdot (a_i - \bar{a}) \cdot f_\theta(x_i),
$$

(6)

where $\alpha \in [0, 1]$ is the auxiliary confidence loss weight and where $\bar{a} = \frac{1}{N} \sum_{i=1}^{N} a_i$ is the base rate of the protected attribute. The first term in equation 6 encourages the weak-to-strong model to make correct predictions while the second term acts as an additional fairness regularizer. The hyperparameter $\lambda_{\text{Fair}}^{\text{W}}$ corresponds to the regularization strength of the weak model while $\lambda_{\text{Fair}}^{\text{WTS}}$ controls the regularization strength for training in this stage.

**Out-of-distribution robustness.** The objective during fine-tuning is to minimize the following loss

$$
\theta^* \in \arg\min_{\theta} \mathcal{L}_{\text{OOD}}(\theta; \lambda_{\text{OOD}}^{\text{W}}, \lambda_{\text{OOD}}^{\text{WTS}}, \alpha, f_w)
$$

$$
= \arg\min_{\theta} \frac{1}{N} \sum_{i=1}^{N} \ell_{\text{WTS-AUX}}\big(x_i, f_\theta(x_i; \lambda_{\text{OOD}}^{\text{WTS}}); \alpha, \lambda_{\text{OOD}}^{\text{W}}, f_w\big),
$$

(7)

where $\alpha \in [0, 1]$ is the auxiliary confidence loss weight. Further, $\lambda_{\text{OOD}}^{\text{W}}$ controls the regularization strength of the fixed weak classifier, while $\lambda_{\text{OOD}}^{\text{WTS}}$ controls the regularization strength of the transfer process. As $\lambda_{\text{OOD}}^{\text{WTS}} = 0$, we are back to our TFT strategy, and as $\lambda_{\text{OOD}}^{\text{WTS}} = \lambda_{\text{OOD}}^{\text{W}} = 0$ the model is trained without any regularization, reverting to the No TFT strategy.

**Adversarial Robustness.** The training objective combines the losses from both clean and adversarial samples:

$$
\theta^* \in \arg\min_{\theta} \mathcal{L}_{\text{Adv}}(\theta; \lambda_{\text{Adv}}^{\text{W}}, \lambda_{\text{Adv}}^{\text{WTS}}, \alpha, f_w)
$$

$$
= \arg\min_{\theta} \frac{1}{N} \sum_{i=1}^{N} (1 - \lambda_{\text{Adv}}^{\text{WTS}}) \, \ell_{\text{WTS-AUX}}(x_i, f_\theta; \alpha, \lambda_{\text{Adv}}^{\text{W}}, f_w) + \lambda_{\text{Adv}}^{\text{WTS}} \, \ell_{\text{WTS-AUX}}(x_i', f_\theta; \alpha, \lambda_{\text{Adv}}^{\text{W}}, f_w),
$$

(8)

where $\lambda_{\text{Adv}}^{\text{W}}$ controls the regularization strength of the fixed weak classifier, while $\lambda_{\text{Adv}}^{\text{WTS}}$ controls the regularization strength of the transfer process. As $\lambda_{\text{Adv}}^{\text{WTS}} = 0$, we are back to our TFT strategy, and as $\lambda_{\text{Adv}}^{\text{WTS}} = \lambda_{\text{Adv}}^{\text{W}} = 0$ the model is trained without any regularization, reverting to the No TFT strategy.

## A.2 CHOOSING THE HYPERPARAMETERS BASED ON TRADE-OFF CURVES

In this section, we provide an illustrative example of how we selected the parameters for the strong baselines, using adversarial robustness as a case study. We plotted trade-off curves between the trustworthiness properties and task performance, selecting the parameter that corresponds to the optimal trade-off in the top right corner of the Figure A1. We set $\lambda_{Adv}$ for the weak and strong model by independently fine-tuning them on training subset and evaluating on the test subset. We plot original task performance vs. adversarial performance for different values of $\lambda_{Adv}$ and pick the value that offers the best trade-off between clean and adversarial accuracy. Figures A1a and A1b show that $\lambda_{Adv} = 0.3$ achieves the best combined accuracies on original and adversarial samples for both models. Fixing $\lambda_{Adv}$ for the weak model to 0.3, we repeat the same analysis for the weak-to-strong model trained with the naive loss function. Figure A1c shows that $\lambda_{Adv} = 0.3$ offers the best trade-off for the weak-to-strong model as well. Fixing the $\lambda_{Adv}$ parameter to 0.3 for the

weak and weak-to-strong models, we vary the $\alpha$ parameter for the auxiliary loss function and plot in figure A1d. We observe that $\alpha = 0.1$ achieves the highest accuracy on both original and adversarial samples. We perform similar analyses for the warm-up period, $\alpha$, and the number of fine-tuning epochs in Figures A1e and A1f. We select the values 0.2 and 6, respectively, for these training parameters.

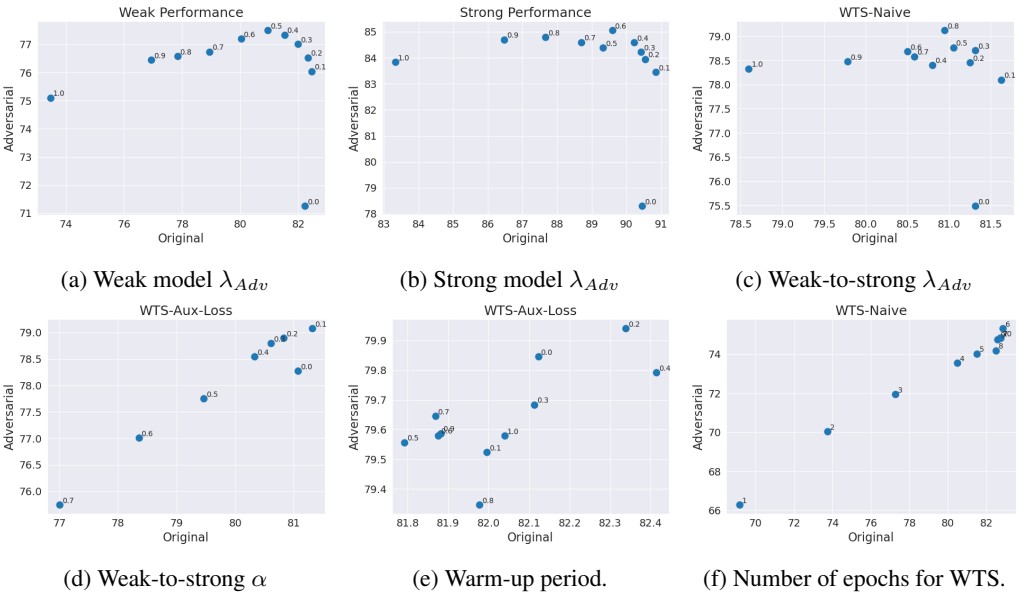

(a) Weak model $\lambda_{Adv}$      (b) Strong model $\lambda_{Adv}$      (c) Weak-to-strong $\lambda_{Adv}$

(d) Weak-to-strong $\alpha$      (e) Warm-up period.      (f) Number of epochs for WTS.

Figure A1: Trade-off between original and adversarial accuracy for different training parameters.

Similarly, for OOD robustness, we set the standard deviation of the Gaussian Noise to $2e - 3$ for both the weak model (Pythia 14M) and the strong model (Pythia 410M). This value was chosen as it allows both models to achieve a balanced trade-off between OOD robustness and task performance. With the noise standard deviation fixed, we conduct trade-off experiments by separately adjusting the maximum $\alpha$ value for auxiliary loss, the warm-up period, and the number of training epochs. For optimal balance between OOD robustness and task performance, these parameters are set to 0.25, 0.2, and 1, respectively.

## B COMPREHENSIVE PLOTS ACROSS STRATEGIES

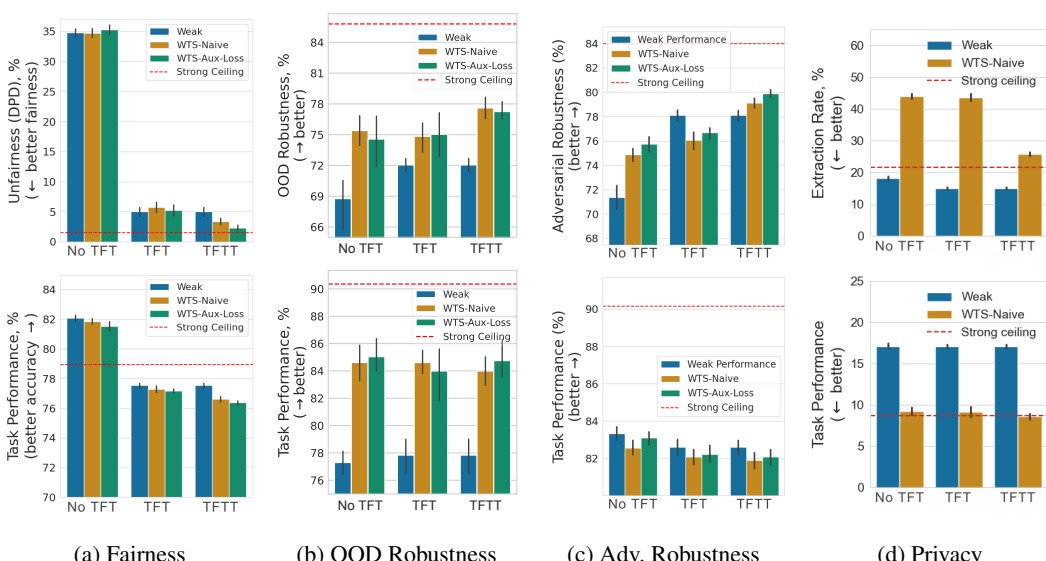

(a) Fairness   (b) OOD Robustness   (c) Adv. Robustness   (d) Privacy

Figure A2: **Weak-to-strong trustworthiness for Pythia 14M/410M models**. Trustworthiness properties and task performance for our four properties: Fairness, OOD Robustness, Adversarial Robustness, and Privacy. Note that lower values are better for the top plot in Figure A2a as the y-axis is Unfairness (DPD). Similarly, lower values are better for the top plot in Figure A2d as the the y-axis is Extraction Rate. Results for WTS-Aux-Loss for privacy are omitted since it was the only task involving free data generation, making the auxiliary loss function inapplicable.

## C DETAILED SENSITIVITY ANALYSIS

In this section, we study the sensitivity of our fine-tuning strategies to key training parameters like $\lambda$ and $\alpha$.

**Impact of Auxiliary Loss Weighting** ($\alpha_{max}$). The auxiliary loss weighting parameter $\alpha_{max}$ (maximum alpha) plays a crucial role in balancing the adherence to the weak model's outputs and the strong model's confidence in its predictions. Higher values of $\alpha_{max}$ place more emphasis on the strong model's own predictions rather than closely following the weak model's outputs. We examine the effect of varying $\alpha_{max}$ from 0 to 1 on the performance of the weak-to-strong models. Our experiments showed a degradation of performance with increasing $\alpha_{max}$. As $\alpha_{max}$ increases from 0 to 1, the performance of the weak-to-strong models trained with the auxiliary loss (WTS-Aux-Loss) tends to worsen. Therefore, selecting an appropriate value of $\alpha_{max}$ is essential to maintain a balance between leveraging the weak model's trustworthiness and allowing the strong model to develop its capabilities. Our results suggest that lower $\alpha_{max}$ values are preferable for effective weak-to-strong trustworthiness transfer. For our models, we chose $\alpha_{max}$ values from 0.1 to 0.4.

**Impact of Larger Models (6.9B).** We show that WTS trustworthiness trends are consistent when scaling up the strong model. As referenced in Section 4.3, Figures A7 to A10, show four different weak/strong model size configurations (14M/410M, 70M/410M, 14M/1B, 70M/1B) with consistent property-specific weak-to-strong trustworthiness trends holding across model sizes. We also extended our model size sensitivity analysis to include Pythia 6.9B as the strong model for fairness, OOD robustness, and adversarial robustness. The 6.9B model required multiple GPUs to train, and DP-SGD currently does not support multi-GPU computations, so we did not provide 6.9B results for privacy. Figure A11 displays the results and demonstrates similar weak-to-strong trustworthiness trends as the previous model configurations. While weak-to-strong trustworthiness is inconsistent at the TFT strategy, we see consistent weak-to-strong trustworthiness at the TFTT strategy.

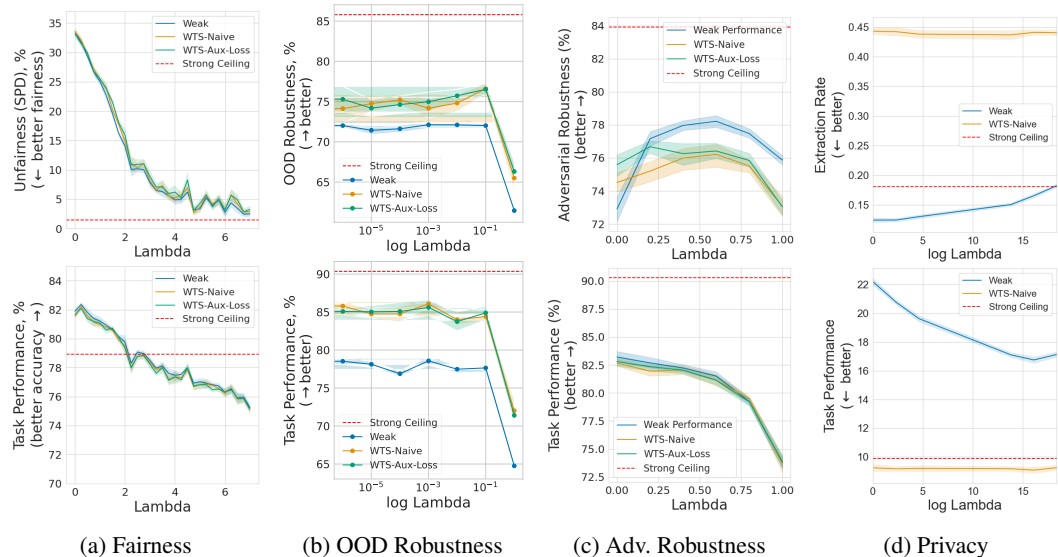

(a) Fairness      (b) OOD Robustness      (c) Adv. Robustness      (d) Privacy

Figure A3: **Full Plot for Varying Lambda for TFT**. Results for WTS-Aux-Loss for privacy are omitted since it was the only task involving free data generation, making the auxiliary loss function inapplicable.

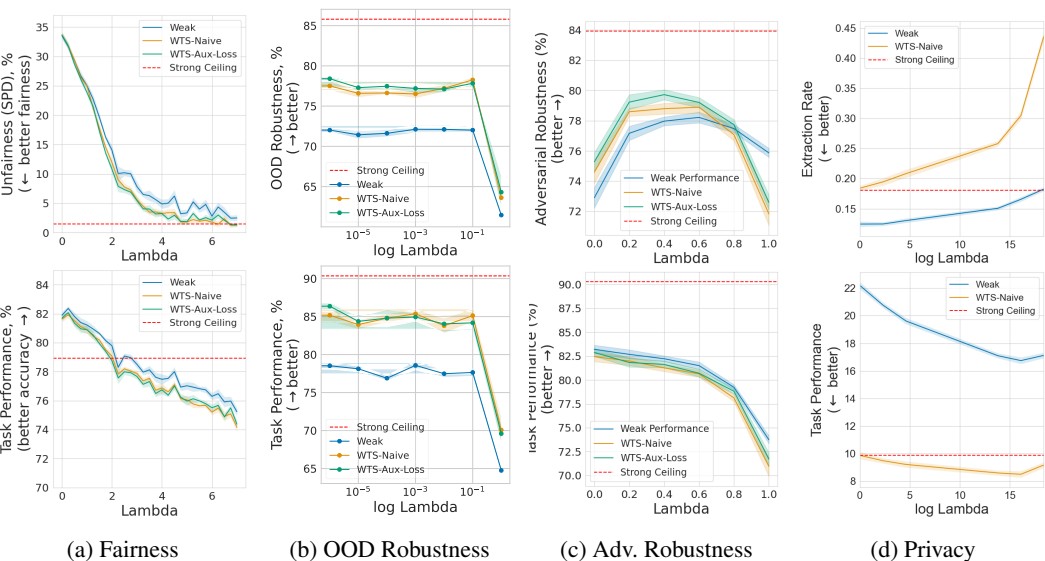

(a) Fairness      (b) OOD Robustness      (c) Adv. Robustness      (d) Privacy

Figure A4: **Full Plot for Varying Lambda for TFTT**. Results for WTS-Aux-Loss for privacy are omitted since it was the only task involving free data generation, making the auxiliary loss function inapplicable.

**Impact of Additional Metrics.** We include multiple trustworthiness definitions to further support the weak-to-strong trustworthiness trends we observed. In Figure A12, we examine an additional fairness metric: equalized odds (true positive rate). The consistent weak-to-strong fairness trend is maintained across both demographic parity and equalized odds. In Figure A13, we examine an additional privacy metric: membership inference attack. We continue to see no weak-to-strong privacy across both extraction and membership inference attacks.

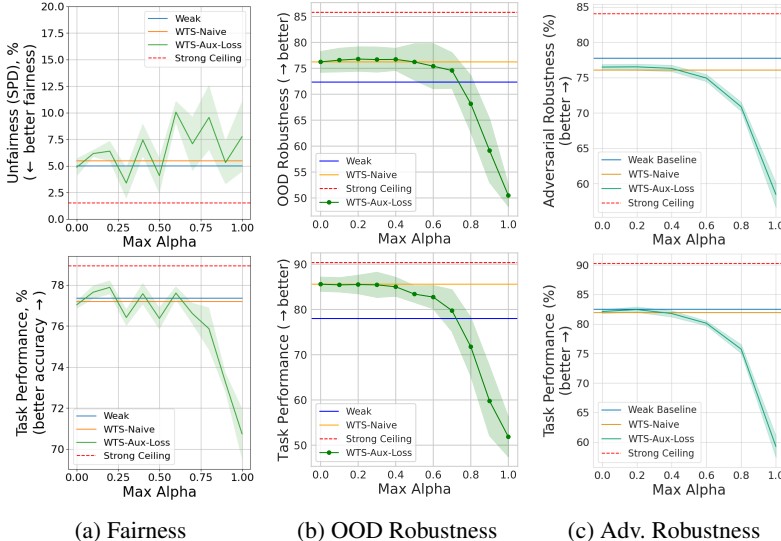

(a) Fairness  (b) OOD Robustness  (c) Adv. Robustness

Figure A5: **Varying Max Alpha for TFT.** Results on privacy are omitted since it was the only task involving free data generation, making the auxiliary loss function inapplicable.

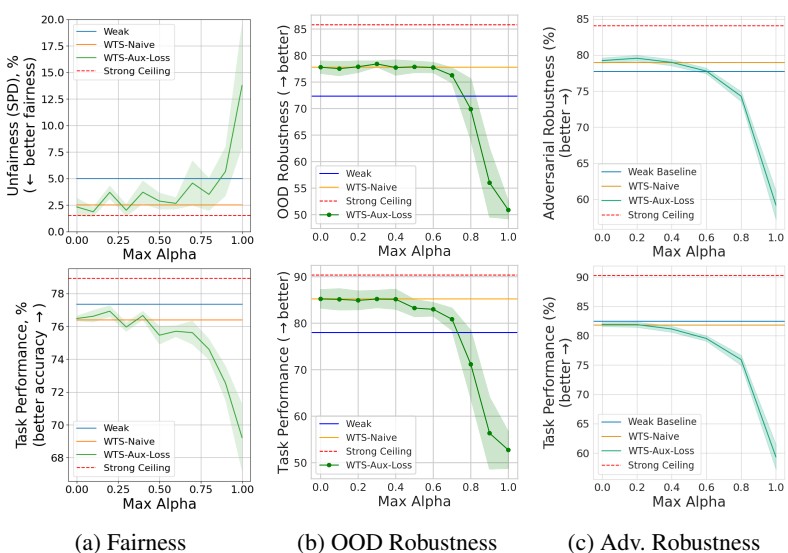

(a) Fairness  (b) OOD Robustness  (c) Adv. Robustness

Figure A6: **Varying Max Alpha for TFTT**. Results for WTS-Aux-Loss for privacy are omitted since it was the only task involving free data generation, making the auxiliary loss function inapplicable.

# D  DATASET AND EVALUATION DETAILS

## D.1  DATASET DETAILS

- **Adult:** The Adult dataset is derived from the 1994 U.S. Census database and contains 48,842 instances with 14 attributes. The task is to classify whether an individual's income exceeds $50K (USD) per year. We selected the "sex" feature as the sensitive attribute to evaluate fairness-related properties. Extraction was done by Barry Becker from the 1994 Census database. Adult dataset has a CC-BY-4.0 license, which we abide by.
- **ACS PUMS Employment:** The Census Bureau's American Community Survey (ACS) Public Use Microdata Sample (PUMS) includes information about U.S. residents' age, sex, race, edu-

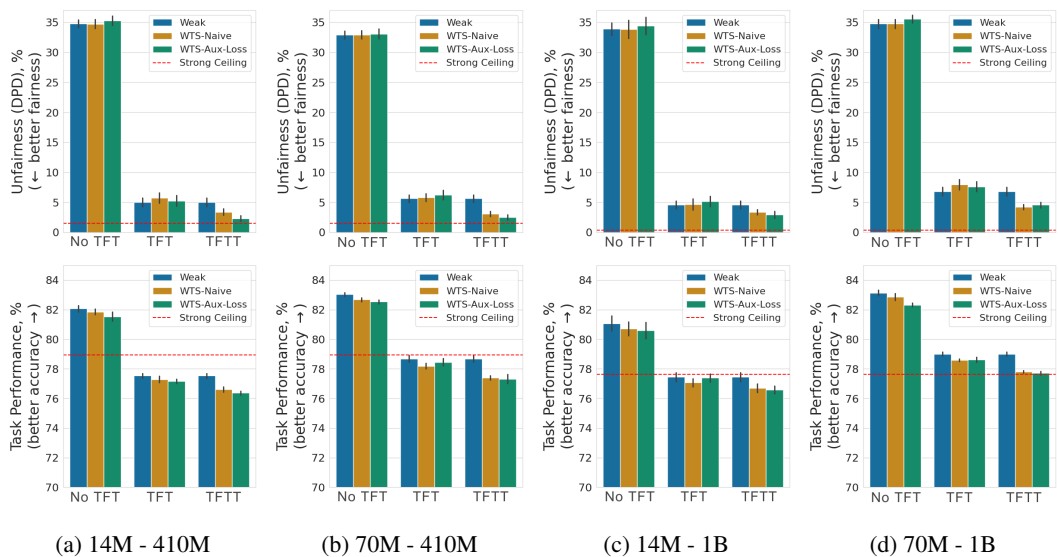

(a) 14M - 410M   (b) 70M - 410M   (c) 14M - 1B   (d) 70M - 1B

Figure A7: **Varying model size for fairness.** Weak-to-strong trustworthiness trends hold for fairness cross multiple model size configurations.

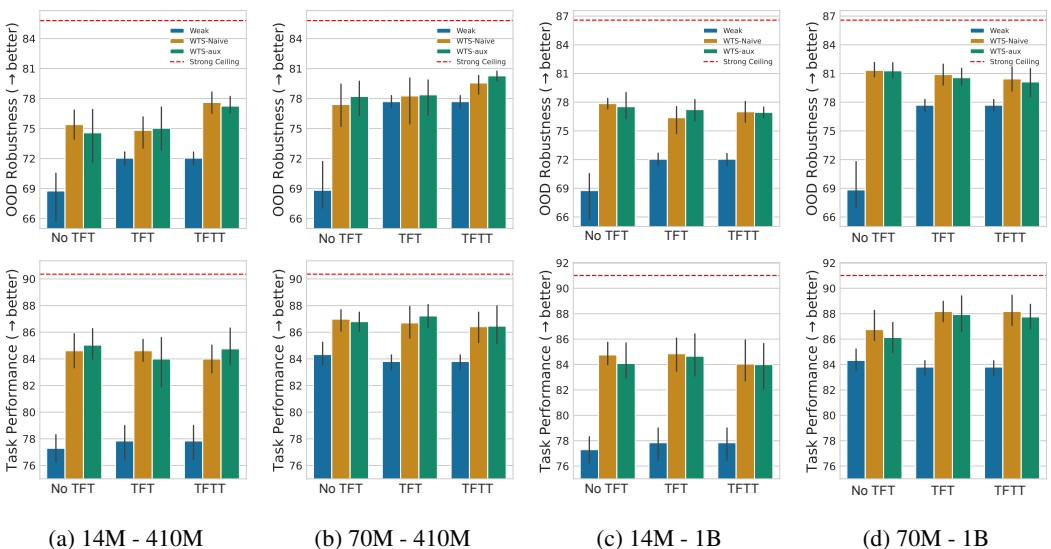

(a) 14M - 410M   (b) 70M - 410M   (c) 14M - 1B   (d) 70M - 1B

Figure A8: **Varying model size for OOD Robustness.** Weak-to-strong trustworthiness trends hold for OOD robustness cross multiple model size configurations.

cation, employment, and other demographics. The task is to classify whether an individual is employed. ACS PUMS dataset has a CC-BY-4.0 license, which we abide by.

- **OOD Style Transfer**: The OOD Style Transfer dataset is based on the SST-2 sentiment classification dataset but incorporates a variety of text and style transformations. The transformations (e.g., shifts in language style, vocabulary, syntax, and tone) are applied at both the word and sentence level while preserving the original meaning (Wang et al., 2023). The task is to correctly classify the sentiment of inputs. OOD Style Transfer dataset has a CC-BY-SA-4.0 license, which we abide by.

- **AdvGLUE++**: AdvGLUE++ is a collection of six datasets contain clean and adversarial input samples for six NLP tasks: Sentiment analysis (SST-2), duplicate question detection (QQP), multi-genre natural language inference (MNLI, MNLI-mm), recognizing textual entailment (RTE), and question answering (QNLI) (Wang et al., 2023). It contains around 2K to 15K sam-

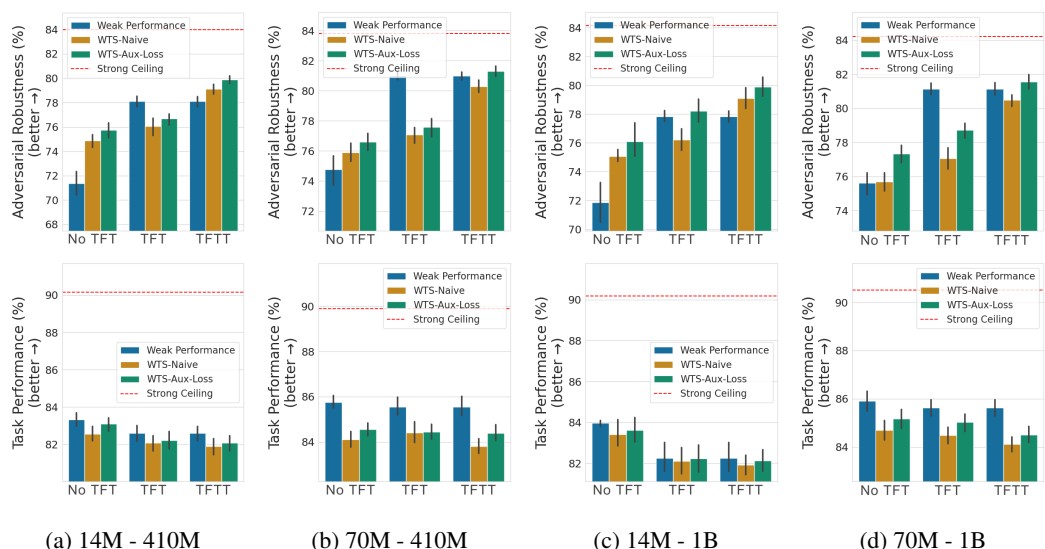

Figure A9: **Varying model size for adversarial robustness.** Weak-to-strong trustworthiness trends hold for adversarial robustness cross multiple model size configurations.

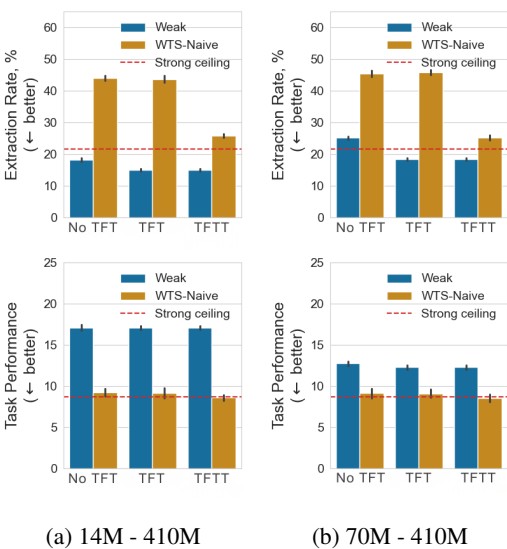

Figure A10: **Varying model size for privacy.** No weak-to-strong trustworthiness trends hold for privacy cross multiple model size configurations. Due to memory limitations of training models with DP-SGD we did not train the 1B or 6.9B models.

ples for each of the six tasks. We randomly sample up to 10K samples for each task and aggregate the performance by averaging over these six tasks. AdvGLUE++ datasets have a CC-BY-SA-4.0 license, which we abide by.

- **Enron Emails**: The Enron Emails dataset contains over 600K emails generated by employees of the Enron Corporation (Klimt & Yang, 2004). it includes sensitive personal information, such as email addresses, phone numbers, credit card numbers, and Social Security Numbers, which could be memorized and extracted by language models. For fine-tuning, we randomly subsampled 10K data points. Enron Emails dataset has a Apache License 2.0, which we abide by.

- **AG News**: The AG News dataset consists of 120,000 training samples and 7,600 test samples of news articles categorized into 4 classes: World, Sports, Business, and Science/Technology. Each sample contains a title and description extracted from AG's news corpus, with balanced distri-

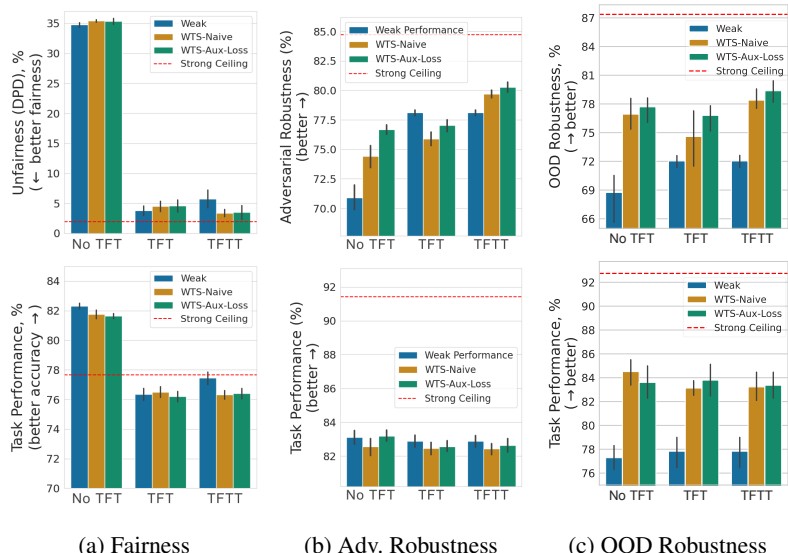

(a) Fairness  (b) Adv. Robustness  (c) OOD Robustness

Figure A11: **Model Size Analysis on Pythia 6.9B**. Results for model size sensitivity with Pythia 14M as the weak model and Pythia 6.9B as the strong model for fairness, adversarial robustness, and OOD robustness properties. We see that the WTS trends we identified earlier are maintained for the larger strong model.

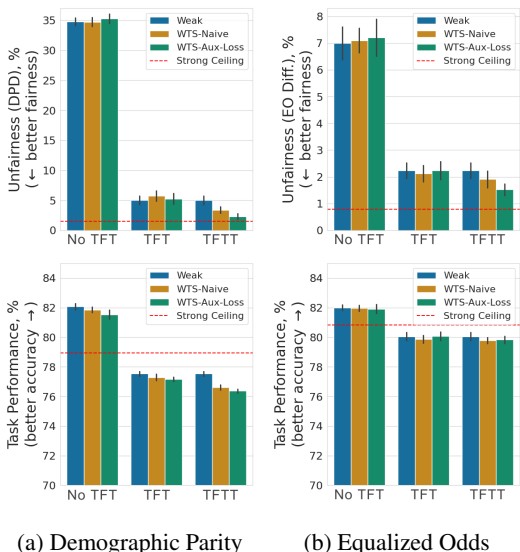

(a) Demographic Parity  (b) Equalized Odds

Figure A12: **Sensitivity to Fairness Metrics**. Side-by-side results for two fairness metrics: Demographic Parity and Equalized Odds (True Positive Rate). The weak-to-strong trustworthiness trends are maintained across both metrics.

bution across classes. AG News data was made by Antonio Gulli (http://groups.di.unipi.it/~gulli/AG_corpus_of_news_articles.html) and permitted for non-commercial use, which we abide by.

## D.2 DATA USAGE DURING TRAINING AND EVALUATION

Figure A15 describes which data is used for training the weak and the weak-to-strong models as well as for evaluating of the weak-to-strong model.

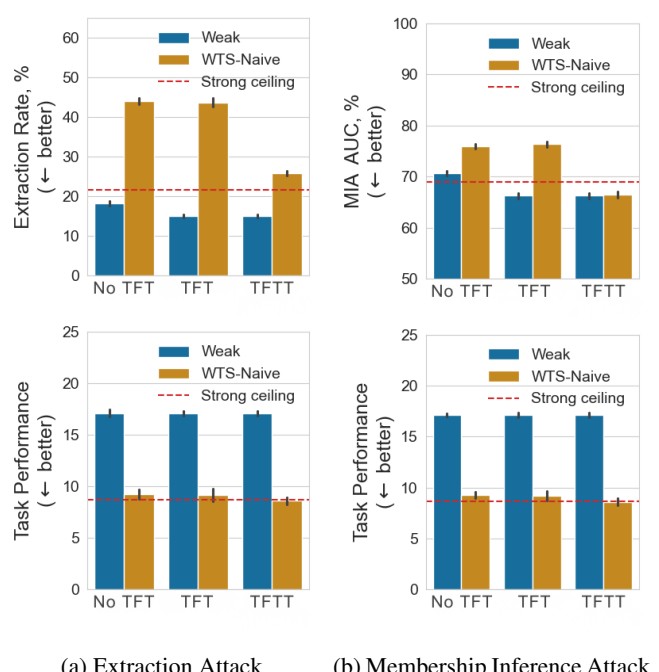

(a) Extraction Attack      (b) Membership Inference Attack

Figure A13: **Sensitivity to Privacy Metrics**. Side-by-side results for two privacy metrics: Extraction Attack and Membership Inference Attack. While TFTT does not achieve weak-to-strong trustworthiness, it still leads to simultaneous improvement of privacy and performance for weak-to-strong models.

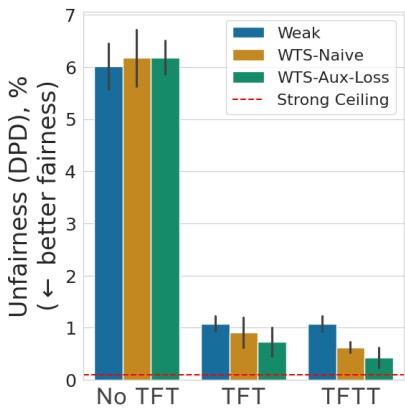

Figure A14: Additional Fairness Dataset: ACS PUMS Employment

**Data used to train the WTS model.** The weak model $f_w$ is trained on the labeled dataset $D_W = \{(x_i, y_i)\}$. Once trained, we use the weak model $f_w$ to label the weak-to-strong learning dataset $D_{WTS} = \{(x_i, y_i)\}$ resulting in $D_{WTS'} = \{(x_i, f_w(x_i))\}$. We use $D_{WTS'}$ to train the weak-to-strong model $f_\theta$. Notably, there is no overlap between $D_{WTS}$ and $D_W$.

**Trustworthiness Evaluation.** We evaluate the trustworthiness properties adversarial robustness, OOD robustness as well as Demographic Parity and Equalized Odds for all models (weak model, weak-to-strong model, and strong ceiling) on the same held out test set for the respective problem. For privacy, we evaluate the trustworthiness properties of the weak and the weak-to-strong model on their training set $D_W$ while the privacy leakage for the WTS model is evaluated on $D_{WTS}$. For privacy considerations, we evaluated the trustworthiness properties of models on their training set $D_W$, while the privacy leakage for the WTS model is assessed on $D_{WTS}$.

Table 2: Additional Privacy Dataset: AG News

| Strategy | Model | Extraction Rate |
|----------|-------|-----------------|
| No TFT | Weak | 0.059 |
| No TFT | WTS-Naive | 0.081 |
| TFT | Weak | 0.050 |
| TFT | WTS-Naive | 0.102 |
| TFTT | Weak | 0.051 |
| TFTT | WTS-Naive | 0.092 |

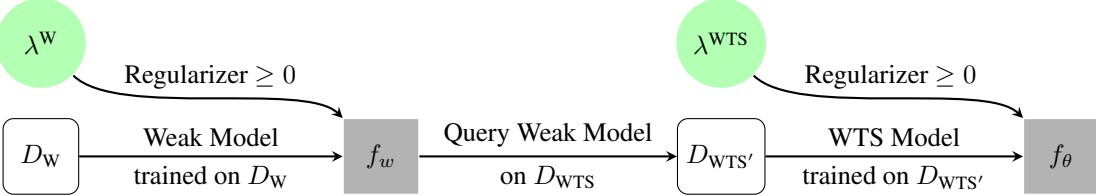

(a) **Model training overview.** The weak model $f_w$ is trained on $D_W = \{(x_i, y_i)\}$. Subsequently, we use the weak model $f_w$ to label the weak-to-strong learning dataset $D_{WTS} = \{(x_i, y_i)\}$ resulting in $D_{WTS'} = \{(x_i, f_w(x_i))\}$. We use $D_{WTS'}$ to train the weak-to-strong model $f_\theta$.

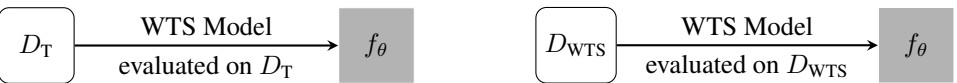

(b) **Trustworthiness property evaluation.** Typically, the trustworthiness properties for the WTS model are evaluated on a separate test set $D_T$.

(c) **Privacy Leakage Evaluation.** The privacy leakage for the WTS model is evaluated using the ground truth train set $D_{WTS}$.

Figure A15: **Data usage during training and evaluation.** In Figure A15a, we describe which data is used to train the weak and the weak-to-strong models, while Figures A15b and A15c describe which data is used for evaluation.

## D.3 ADDITIONAL ADVERSARIAL ROBUSTNESS DATASET DETAILS

We create training, holdout and test subsets of the AdvGLUE++ dataset using 40%, 40% and 20% of samples, respectively, from each task in the dataset. We use the training subset to fine-tune our models to be adversarially robust. We use the holdout subset to generate labels from the weak model to be used in the weak-to-strong learning process. To evaluate the clean and adversarial accuracy of our models, we evaluate them on a test subset of the AdvGLUE++ dataset and average the performance across the six NLP tasks in this dataset.

In particular, to evaluate weak-to-strong trends in adversarial robustness, we use the AdvGLUE++ dataset (Wang et al., 2023), an extension of the AdvGLUE dataset (Wang et al., 2021). AdvGLUE++ is a comprehensive benchmark designed to test adversarial robustness across multiple natural language processing (NLP) tasks and adversarial attack algorithms. This dataset includes adversarial examples for six widely used NLP tasks, each representing a distinct domain or linguistic challenge. The Stanford Sentiment Treebank (SST-2) task involves sentiment analysis, requiring the classification of sentences as having a positive or negative sentiment. The Quora Question Pairs (QQP) task identifies whether two questions convey the same meaning. The Multi-Genre Natural Language Inference (MNLI) task requires reasoning about entailment, contradiction, or neutrality between pairs of sentences. It includes a mismatched variant, MNLI-mm, where validation and test data originate from out-of-domain sources, increasing the challenge of generalization. The Question-answering NLI (QNLI) task is framed as an entailment problem between a question and an answer candidate. The Recognizing Textual Entailment (RTE) is a binary entailment task that aims to determine whether the meaning of one text can be inferred from another.

Adversarial examples in AdvGLUE++ are generated using a variety of attack algorithms, each representing a distinct perturbation strategy. TextBugger introduces typo-based perturbations that minimally alter characters while preserving the utility of benign text. TextFooler generates embedding similarity-based perturbations by substituting words with contextually plausible alternatives. BERT-ATTACK leverages BERT's language modeling capabilities to create context-aware adversarial samples. SememePSO relies on semantic representations and combinatorial optimization to generate knowledge-guided perturbations. SemAttack employs semantic optimization-based techniques by manipulating various semantic spaces to produce natural-looking adversarial texts.

The experimental results for adversarial robustness are presented as aggregated accuracy values across all six tasks and five attack algorithms. This approach enables us to evaluate the weak-to-strong trends in a comprehensive and robust manner. The results show that our findings are consistent across a wide range of NLP tasks and adversarial attacks, indicating that they are not influenced by the specific characteristics of any single setting.

### D.4    ADDITIONAL OOD DATASET DETAILS

We use the same OOD data created by Wang et al. (2023). For ID data, we use the original SST-2 dataset but exclude the samples that are source samples for creating the OOD data. We split the ID data into training, validation, and heldout subsets. Specifically, 50% of the ID data is allocated for training and validation, where 95% of that portion is used for training and the remaining 5% is for validation. The other half represents the held-out data that is used for generating labels from the weak model for weak-to-strong fine-tuning. For evaluation, we use the in-distribution validation samples to measure ID performance and the OOD test samples to obtain OOD performance.

# E OVERVIEW TABLE

Table 3: **Overview table.** Trustworthiness properties, their corresponding metrics, datasets used, and tasks performed on each dataset.

| Property | Metrics | Datasets | Tasks |
|---|---|---|---|
| Fairness | • Demographic Parity
• Equalized Odds | • Adult
• ACS PUMS | • Income classification with "sex" as the sensitive attribute |
| OOD Robustness | • Robust Accuracy (RA) on OOD test data | • OOD Style Transfer: a collection of 10 datasets with different text and style transformations (based on the SST-2 dataset) | • Sentiment classification on 10 different text and style transformations |
| Adversarial Robustness | • Robust Accuracy (RA) on adversarial test data | • AdvGLUE++: a collection of six datasets
1. SST-2
2. QQP
3. MNLI
4. MNLI-mm
5. RTE
6. QNLI | • Sentiment analysis
• Duplicate question detection
• Multi-genre natural language inference
• Recognizing textual entailment
• Question answering |
| Privacy | • Extraction attack
• Membership inference attack | • Enron Emails
• AG-News | • Sensitive data leakage detection |

# F EXPERIMENTAL DETAILS

**Models:** We use the Pythia models from EleutherAI (Biderman et al., 2023). They have a Apache License 2.0, which we abide by.

**Statistical Significance:** We report 1 standard deviations for our experiments over multiple trials (10 for fairness, 15 for OOD robustness, 15 for adversarial robustness, 3 for privacy).

**Compute:** Each experiment was run on 1 NVIDIA A100 80GB GPU on an internal cluster.

Table 4: Hyperparameters

| Hyperparameter | Fairness | OOD Robustness | Adversarial Robustness | Privacy |
|---|---|---|---|---|
| Epochs | 5 | 1 | 6 | 1 |
| Learning rate | 5e-5 | 1e-5 | 1e-5 | 5e-5 |
| Optimizer | AdamW | AdamW | AdamW | Adam |
| Lambda | 4.25 | 0.002 | 0.3 | 1e6 |
| Alpha | 0.3 | 0.2 | 0.1 | N/A |

