# OpenReview forum: "Can Trustworthiness Generalize? Leveraging Weak Supervision for Stronger Models"
_ICLR.cc/2026/Conference — ICLR 2026 Conference Withdrawn Submission_

### Official Review · Reviewer_Znbd · 2025-10-15

**Soundness:** 3
**Presentation:** 3
**Contribution:** 1
**Rating:** 2
**Confidence:** 4

**Summary:**

This paper introduces the novel paradigm of **weak-to-strong trustworthiness** (inspired by weak-to-strong generalization) to address the critical challenge of enhancing trustworthiness in Large Language Models (LLMs) when real-world ground truth labels are unavailable. The core approach involves fine-tuning a stronger model on labels generated by a weaker model, augmented with specific regularization strategies. Experimental results indicate that properties such as fairness and robustness can successfully generalize and improve, although privacy does not exhibit the same trend.

**Strengths:**

* **Paradigm:** The proposal of weak-to-strong trustworthiness as a mechanism to enhance LLM trustworthiness, motivated by the weak-to-strong generalization literature, appears to be interesting.
* **Comprehensive Scope:** The study is commendable for considering and experimentally evaluating several critical trustworthiness dimensions (fairness, robustness, privacy).
* **Clarity and Quality:** The paper is well-written and clearly presented.
* **Dataset Consideration:** The use of the updated Adult dataset (Ding et al., 2021) is a thoughtful choice, addressing known limitations and issues associated with the standard version frequently used in fairness research.
* **Notation Detail:** The conscious effort to avoid notation confusion between Demographic Parity (DP) and Differentially Privacy (DP) by modifying the notation is a small but appreciated detail that contributes to clarity.

**Weaknesses:**

## 1. Summary of Weaknesses
I appreciate the authors' efforts on this paper and a potential rebuttal/revision. My primary goal with this review is to offer constructive suggestions from my perspective, and I'm certainly willing to increase my score if my concerns are properly addressed in the revision.

My primary concern centers on the problem setting and its limited scope, as many of my subsequent criticisms stem from this issue. Firstly, in my understanding, the paper's core objective is to enhance trustworthiness under imperfect supervision, as stated in Lines 40-44:

> As a result, the labels used for training are noisy or incomplete, rather than perfect ground truth. The challenge of imperfect supervision parallels a question in AI alignment: if we only have access to potentially biased supervision (like imperfect human decisions), how can we control more capable AI systems to be more aligned with human values and trustworthiness?

This research, which investigates whether a stronger student model fine-tuned on a weaker teacher model's labels can surpass the teacher's performance in trustworthiness properties, faces a potential novelty concern. Specifically, if the core contribution is merely an experimental demonstration that weak-to-strong generalization also holds true for certain trustworthiness-related tasks, the work may be viewed as primarily confirmatory rather than proposing a fundamentally new mechanism or insight.

## 2. Related Work

The related work section is currently insufficient. While it covers the trustworthiness literature adequately, it lacks discussion on weak-to-strong generalization. Given that the paper's core motivation and paradigm are directly drawn from this field, a dedicated review of weak-to-strong generalization literature is essential. I strongly encourage the authors to incorporate this material, potentially in the Appendix if space is severely constrained.

## 3. Fairness

* **Regularization Appropriateness:** The paper utilizes the covariance regularization method from Zafar et al. (2017). A key benefit of that work was the use of a linear classifier resulting in a convex constrained optimization problem, which is numerically tractable. This is fundamentally different from the setting in this paper, which uses complex LLMs and potentially non-linear boundaries on sophisticated datasets. The Adult dataset, in particular, is often well-solved by simple classifiers, which makes it less convincing as a representative dataset for the complexities inherent in LLM fairness research.
* **Evaluation Metric:** The evaluation of fairness should not discard accuracy. A superior algorithm in the fairness literature is one that improves the fairness-accuracy trade-off (achieving better fairness for a given accuracy, or vice-versa). Simply imposing stronger regularization leads to better fairness at the expense of lower accuracy. The results should simultaneously consider both fairness and accuracy to properly situate the proposed method's utility, in line with established evaluation practices (e.g., Zafar et al. (2017) and related works).

## 4. Adversarial Robustness

* **Adversarial Training Context:** The current approach of training models on adversarial examples is essentially Adversarial Training. The paper should explicitly cite the original or foundational papers on adversarial training for proper context.
* **Advanced Attacks:** While using AdvGLUE++ is positive, I question whether simply using datasets to represent a trustworthiness dimension is sufficient. I suggest investigating the utility of evaluating against more advanced, state-of-the-art attack methods designed specifically for LLMs. The authors can provide justification for their current experimental scope.
* **Theoretical Connection:** Adversarial robustness is fundamentally related to individual fairness, one of the prominent definitions within the fairness community. A brief discussion exploring this connection would significantly enhance the paper's depth. Furthermore, there are theoretical works discussing the broader relationships among fairness, robustness, and privacy. Including a short, focused discussion (possibly in the Appendix) on these theoretical connections would further strengthen the paper.

## 5. Experiments

**Baseline.** The paper's core motivation (Lines 41-44) is to improve LLM trustworthiness under imperfect/biased supervision. This setting closely aligns with problems studied in the literature on weakly-supervised learning, self-supervised learning, noisy label learning, and knowledge distillation in the context of trustworthiness (fairness, robustness, privacy). For example, the related field of "Fair weak-supervised learning" is not even mentioned. To validate the proposed paradigm, the experiments should ideally include existing weak-supervised learning / knowledge distillation techniques tailored for improving trustworthiness as baselines. Without these, it is difficult to ascertain whether weak-to-strong trustworthiness is a uniquely effective approach or if its benefits are partially captured by existing methods operating under similar imperfect-label conditions.

**Questions:**

1.  What is the fundamental difference between your experimental setting and existing weak-supervised learning techniques?
2.  Do you believe more advanced fairness datasets and fine-tuning strategies should be employed in your experiments to better reflect the complexities of modern LLMs?
3.  Do you think some advanced attack methods should be incorporated into your experiments on adversarial robustness to provide a more rigorous evaluation?
4.  Why did you not consider incorporating baselines from the weak-supervised learning / knowledge distillation literature concerning fairness, robustness, or privacy?

---

### Official Review · Reviewer_N5Br · 2025-10-30

**Soundness:** 2
**Presentation:** 2
**Contribution:** 2
**Rating:** 4
**Confidence:** 3

**Summary:**

This paper introduces a concept, weak-to-strong trustworthiness, to investigate whether a strong model can become more trustworthy than a weak model by only fine-tuning on its labels. The authors propose two new strategies, TFT, which regularizes weak model training, and TFTT, which regularizes both weak model training and WTS learning, to improve WTS trustworthiness. Through experiments, they find that standard WTS and TFT are insufficient for trustworthiness generalization. However, the TFTT strategy successfully generalizes fairness and robustness. The paper also concludes that privacy is a fundamental exception due to its inherent conflict with model capacity.

**Strengths:**

+ New and an important topic
+ Comprehensive experimental evaluation
+ TFTT strategy shows strong positive results and is robust to hyperparameter choices

**Weaknesses:**

- Analysis of the results of the TFT strategy is insufficient
- Results for TFT (OOD robustness) appear to be misinterpreted
- A key methodological limitation (WTS-Aux-Loss inapplicability) is not reported in the main text

**Questions:**

1. The conclusion that "privacy cannot be generalized" is based on a flawed experimental premise. In your own experiments (e.g., Figure 6), the "Strong Ceiling" (the strong model's best privacy performance) is significantly worse than the weak model's privacy performance. If the strong model's theoretical upper bound is already inferior to the weak model, then "weak-to-strong" generalization (where the strong model surpasses the weak one) is logically impossible by definition. This experiment does not prove that privacy "failed to generalize". It proves that this specific metric is fundamentally opposed to model scaling. We suggest reframing the conclusion to address this logical conflict.

2. The analysis of the TFT strategy is a significant weakness. The paper states that TFT failed for fairness and adversarial robustness, but fails to analyze why. Furthermore, the claim that TFT did show generalization for OOD robustness (Figure 4b), but it may appear to be a misreading of the data. The WTS-Naive model under the TFT strategy (lower than 75% robustness) shows no improvement over the WTS-Naive model under the No TFT strategy (over 75% robustness). This strongly suggests that the 75% robustness is merely the baseline benefit of model scale, and that the TFT strategy also failed for OOD robustness. We recommend revisiting this part of the analysis.

3. There is a lack of reporting transparency. WTS-Aux-Loss is a core WTS transfer method in your evaluation, yet it is omitted from all privacy-related experiments. The only explanation is hidden in the caption of Figure A2 in the appendix, stating it is "inapplicable" due to "free data generation". This is a major methodological limitation for one of your four core properties and must be explicitly stated and discussed in the main body of the paper (e.g., in Section 4.1 or 4.2 ).

---

### Official Review · Reviewer_Y6KT · 2025-10-31

**Soundness:** 2
**Presentation:** 3
**Contribution:** 2
**Rating:** 4
**Confidence:** 3

**Summary:**

To examine if a stronger model can enhance trustworthiness when fine-tuned on a weaker model’s labels, this paper introduces two fundamental fine-tuning strategies (TFT and TFTT) that leverage trustworthiness regularization during the fine-tuning of the weak model and the weak-to-strong transfer. Experiments show that while some trustworthiness properties, such as fairness, adversarial robustness, and OOD robustness, show improvement in trustworthiness generalization, others like privacy do not exhibit signs of weak-to-strong trustworthiness. This work reveals the potential of weak-to-strong trustworthiness.

**Strengths:**

- The paper is well-structured, and the content is presented logically, making it easy to follow.
- The experiments validate the effectiveness of the proposed methods from various aspects.

**Weaknesses:**

- Although the experiments are extensive, they are limited to Pythia models ranging from 14M to 6.9B. Generalization across different model families is not validated, and it remains unanswered whether the results from small-scale models are applicable to much larger models.
- The paper's core idea is based on WTS (Weak-to-Strong) generalization, which limits its novelty.
- While the TFTT strategy is effective, it requires training in two independent stages, introducing additional complexity and computational overhead. Furthermore, the selection of hyperparameters during the training process across different trustworthiness dimensions and models requires extensive experimental validation, limiting its practical use.
- Capability and trustworthiness are inextricably linked. The paper does not discuss how model capability changes when generalizing trustworthiness, or how trustworthiness is affected when generalizing capability.
- Key training details are missing, such as the specific training and testing data used. The paper also fails to verify generalization between different datasets within the same trustworthiness dimension (e.g., validating between the Adult and ACS PUMS datasets for fairness).
- The potential trade-offs between different trustworthiness dimensions are not discussed.

**Questions:**

See weaknesses.

---

### Official Review · Reviewer_g63y · 2025-11-01

**Soundness:** 2
**Presentation:** 3
**Contribution:** 2
**Rating:** 4
**Confidence:** 4

**Summary:**

The paper investigates an interesting research problem that explores the potential of weak-to-strong trustworthiness. The main motivation comes from the recent “weak-to-strong” paper, and this paper aims to focus on the trustworthiness-related discussions in the weak-to-strong training paradigm. To this end, the paper compares three potential training approaches: 1) no trustworthiness fine-tuning (No TFT), 2) trustworthiness fine-tuning (TFT), and 3) trustworthiness fine-tuning and transfer (TFTT). The experiments contain a wide range of datasets, tasks, and models, which enrich the main observations of the paper.

**Strengths:**

- The research covers various aspects of AI trustworthiness, including fairness, out-of-distribution (OOD) robustness, adversarial robustness, and privacy, enriching the overall discussions in these areas.
- Experimental results are supported by a wide range of datasets, tasks, and models, which improves the credibility of the observations.

**Weaknesses:**

A main concern is whether the “weak-to-strong trustworthiness” framework is the correct lens for interpreting the paper’s observations. It remains unclear if the observed improvements come from the weak model’s signals or simply from the regularization applied during transfer learning. Here are more details:
- It remains unclear if the trustworthiness improvements come from the inherent trustworthiness of the weak model (weak labels) or from the regularization applied during the transfer learning phase. A full ablation study (e.g., including training only with trustworthiness regularizers during transfer without weak model training) would be helpful in analyzing their separate impacts.
- If the regularization applied during the strong model training is the main factor of improvement, then the problem could be re-projected to existing AI trustworthiness research. For example, a fairness improvement in the strong model due to transfer learning regularization might be comparable to outcomes/discussions achieved by traditional fairness algorithms. This suspicion is reinforced by observations that solely applying TFT (i.e., making the weak model more trustworthy) does not consistently yield improvements in fairness and adversarial robustness, as in Figure 4.

**Questions:**

The concerns described in the weaknesses section also cover main questions.

---

### Note · Authors · 2025-11-26

**Comment:**

We thank the reviewers for their time in providing us with valuable feedback. We retract our submission and will revise our submission accordingly.

**Withdrawal Confirmation:**

I have read and agree with the venue's withdrawal policy on behalf of myself and my co-authors.